# Refractive Index Dependence of Fluorescence Enhancement in a Nanostructured Plasmonic Grating

**DOI:** 10.3390/ma16031289

**Published:** 2023-02-02

**Authors:** Margherita Angelini, Eliana Manobianco, Paola Pellacani, Francesco Floris, Franco Marabelli

**Affiliations:** 1Department of Physics, University of Pavia, Via Bassi 6, 27100 Pavia, Italy; 2Plasmore S.r.l, Via Vittorio Emanuele II 4, 27100 Pavia, Italy

**Keywords:** fluorescence, surface plasmon resonance, electric field coupling mechanism, optical characterization, FDTD simulation

## Abstract

Plasmonic gratings are attracting huge interest in the context of realizing sensors based on surface-enhanced fluorescence. The grating features control the plasmonic modes and consequently have a strong effect on the fluorescence response. Within this framework, we focused on the use of a buffer solution flowing across the grating active surface to mimic a real measurement. The refractive index of the surrounding medium is therefore altered, with a consequent modification of the resonance conditions. The result is a shift in the spectral features of the fluorescence emission accompanied by a reshaping of the fluorescence emission in terms of spectral weight and direction.

## 1. Introduction

Plasmonic gratings are one of the main tools used to amplify the fluorescent emission of molecules, which leads to surface-enhanced fluorescence (SEF) [1,2,3,4,5,6,7,8,9]. Additionally, dielectric metastructures based on SEF (e.g., via Bloch surface waves) [10,11,12,13] represent a valid solution. Other possibilities are offered by metallic nanoparticles [14,15,16,17,18,19,20,21] or photonic systems, such as resonant cavities [22,23,24], which are used to exploit the strong field enhancement occurring in their proximity due to localized modes. With respect to localized mode-based systems, SEF-based systems have the advantage of manipulating the emission geometry and, in this scenario, plasmonic gratings give the additional advantage of bringing together, in one single platform, the coupler and the amplifier.

In a recent study [25], we analyzed the plasmonic modes and fluorescence enhancement coupling mechanism in relation to a plasmonic grating, proving that the fluorescence spectral shape depends on the plasmonic field distribution together with the enhancement of its signal. Additionally, the spectral region characterized by the most significative relative fluorescence enhancement is tightly linked to the most peculiar signatures of the plasmonic modes.

For practical applications, it is often required to have buffer solutions flow on the active surface. This implies a change in the refractive index of the surrounding medium and the consequent modification of the resonance conditions also affecting the enhancement intensity [26]. With this in mind, we aimed to extend the previous analysis by studying the reshaping of the fluorescence signal of the ATTO700 dye [27], and the consequent enhancement, induced by the exposure of the plasmonic grating to a solution of 2% ethanol (EtOH) in water. EtOH was added to prevent possible effects due to hydrophobicity. From the point of view of our investigation, this means a change in the refractive index of the surrounding medium from n_AIR_ = 1.000, where the surrounding medium is air, to a value of n_EtOH_ = 1.334, where the surrounding medium is a solution of 2% EtOH in water [28,29].

The following plasmonic field distributions and the corresponding shifts were also considered to confirm and strengthen the findings reported in [25]. The updated electric field expansions are also reported, proving the near-field capability in affecting the fluorescence reshaping.

## 2. Materials and Methods

The plasmonic grating under study was fabricated using a well-assessed technique based on colloidal nanolithography [30]. The nanofabrication process comprises several different steps. At first, poly(methyl methacrylate) (PMMA) is dissolved in anisole and then, in order to form a layer of controlled thickness, is deposited by spin-coating on a commercial SiO_2_ substrate. Secondly, the sample is annealed at a high temperature (180 °C) for 10 min in order to properly remove any residual solvent and simultaneously harden the deposited PMMA layer.

In a separate step, commercial monodisperse spherical beads made of polystyrene with a nominal diameter of 500 nm are deposited on the water surface level of a rectangular Langmuir–Blodgett trough. As a consequence, the polystyrene spherical beads arrange themselves in hexagonal close-packed arrays and thus create a bidimensional crystal with a lattice spacing determined by the beads’ diameter. At this point, the sample composed by the PMMA monolayer on the SiO_2_ substrate is dip-coated in the trough, enabling the transfer of the lattice of colloidal beads on the sample surface. In this way, the PMMA layer is exploited as a sacrificial mask in the reactive O_2_ plasma etching exposition process.

For this reason, by controlling the etching parameters, it is possible to tune the dimension of the beads’ diameter and thus fabricate a pattern with the desired structural features.

After this etching step, the substrate sample is at first coated by a layer of titanium a few nanometers thick, which is deposited directly onto the etched PMMA and polystyrene colloidal mask. This metallic layer is needed to ensure the optimal adhesion of the subsequent deposition of the thicker layer of gold (approximately a few hundred nanometers). In fact, in the following step, by controlling the physical vapor deposition parameters, a gold film of defined thickness is deposited on the sample surface, filling the wells formed in the PMMA layer exposed to reactive ion etching. Once the grating is metal coated, the polystyrene beads are removed by keeping the sample in an ultrasonic bath for a few minutes. Eventually, to enable the rearrangement of the plasmonic crystal structure, the sample substrate is annealed at high temperatures. The resulting sample consists of hexagonally arranged PMMA nanopillars embedded in a gold film on a SiO_2_ substrate. For the sake of simplicity, in the manuscript, the active surface corresponding to the PMMA/Au/Surrounding medium (See Figure 1) interface will be referred to as the front side (FS), and the Au/SiO_2_ as the back side (BS). The ATTO700 dye [27] was selected to provide a proper overlap with the plasmonic modes due to its emission spectrum spanning from 660 to 860 nm. The dye was drop-casted on the FS of the grating and on a bare SiO_2_ glass slide for reference, following the procedure described in [25].

Two polyelectrolytes (PEL), poly(styrene) sulfonate and poly(diallyldimethylammonium) chloride, were used to create a convenient and stable substrate for the dye deposition. Since these polymers carry positive or negative charges, it is possible to build up a nanometric layer with a controlled thickness in terms of the number of alternating charged layers, guaranteeing a well-defined spacer layer between the dye and the metallic surface. Moreover, since the selected dye is a charged molecule [27], the presence of a layer with a proper electrostatic charge improves the uniformity in terms of the dye deposition.

In detail, by alternating positively and negatively charged PEL, five different layers were deposited on both the grating FS and the bare SiO_2_ reference. The PEL coating procedure is composed of several steps. At first, the sample is immersed for two minutes in the selected PEL solution and subsequently with MilliQ water. After the sample is dried under nitrogen flow, the procedure is repeated by alternatively dipping the sample in the two PEL solutions, poly(styrene) sulfonate at 2% and poly(diallyldimethylammonium) chloride at 2%. In this way, the PEL layers are stacked on the sample surface. After reaching 10 PEL layers of 2 nm thick each, the ATTO700 dye was drop-casted on the two samples in a 10 μM aqueous solution.

The sample was optically characterized with broadband (500 to 1110 nm) reflectance (R) and transmittance (T) measurements performed with a commercial Fourier transform spectrometer Bruker IFS66S coupled with a homemade micro reflectometer [25].

Thanks to the momentum conservation guaranteed by the periodicity of the metallic surface, surface plasmons can be directly excited and their signature can be detected in the R and T spectral features [25,30]. A Glan Taylor polarizer was used to select the TE or TM light polarization (electric or magnetic field perpendicular to the plane of incidence, respectively). Both R and T were measured by illuminating the sample from the BS, which consequently kept the FS, i.e., the active surface, clear. This helped to avoid any undesired perturbation to be detected as a result of any variation in the refractive index of the surrounding medium. Operatively, we resorted to a commercial microfluidic cell to properly wet the FS.

Fluorescence was excited using a He-Ne laser, pumping at 632.81 nm, with a power of 75 µW, and collected in epifluorescence geometry using a Labram Dilor spectrometer equipped with an Olympus HS BX40 microscope. The selected objective was characterized by a numerical aperture of 0.25, resulting in a circular excitation spot of 100 µm^2^. An optical density 2 filter was used. With the active surface exposed to air, the fluorescence signals were excited and collected by focusing the excitation spot at the Au layer with both the FS and BS facing the microscope objective. The same procedure was followed for the SiO_2_ reference [25].

Concerning the fluorescence measurements performed with the 2% EtOH solution, the procedure was slightly modified. The solution was drop-casted on the active surface of the grating and the dye layer on the SiO_2_ reference. A thin glass slide was placed on the liquid to guarantee the correct focus of the excitation spot.

A built-in CCD Peltier-cooled camera was used to collect the fluorescence signals, with the possibility to select the spectral range in the acquisition. For the measurements with air as the surrounding medium, the fluorescence signals were collected in the wavelength range from 650 to 820 nm. When the grating active surface was exposed to the 2% EtOH solution, the fluorescence signal collection was performed in the spectral range from 650 to 880 nm.

As far as the computational analysis is concerned, the finite-difference time-domain (FDTD) model successfully proposed in [25] was implemented in Ansys Lumerical FDTD [31] with several additional design steps. As the sample under investigation was characterized by a marginally different optical response in air and considering the additional co-optimization for the 2% EtOH solution surrounding medium, the FDTD simulations required a forward rigorous optimization process. Additional geometrical features were considered to provide a proper view of the electromagnetic behavior that was able to support the refractive index dependence of fluorescence enhancement.

The FDTD model was first built by considering nanofabrication information (the Au layer thickness) together with SEM images of the sample’s active surface (R1 and R3 in Figure 1). Then, the PMMA nanopillar geometrical parameters were retrieved through the optimization of the optical response in terms of R and T spectral shape. In detail, several sweeps on the hidden parameters (R2 and R4 in Figure 1) were performed to tune the spectral position of the different plasmonic features for both n_AIR_ and n_EtOH_ while simultaneously considering the spectral shift corresponding to the refractive index change in the surrounding medium. As a rule of thumb, the pitch value determines the spectral position of the plasmonic modes, whereas the radii of the pillar mainly affect their shape.

The gold layer optical properties were set as the built-in Johnson and Christy data in the Ansys Lumerical [31] material database. The PMMA and the SiO_2_ structures were modeled as perfect dielectric materials by selecting a refractive index of 1.48 and 1.50, respectively. As for air as the surrounding medium, the FDTD background index was fixed as n_AIR_, and the corresponding material portion of air in the nanopillar (see Figure 1) was set as a perfect dielectric material with the same refractive index. An analogous procedure was carried out when the refractive index of the surrounding medium was changed to n_EtOH_.

A plane wave source was placed below the SiO_2_ substrate, illuminating the sample at normal incidence from the BS, in the spectral interval spanning 500 to 1100 nm. The source was linearly polarized, with the electric field oscillating along the x-direction. Two frequency-domain field and power monitors were placed in the FDTD simulation to collect the R (behind the source) and T (above the FS) signals, with these having the same spectral interval as the source. A auto non-uniform mesh type with a mesh accuracy of five was selected after convergence testing. A mesh override of 2.5 nm along the x-, y-, and z-directions was added in the region of the plasmonic grating. Mesh refinement was chosen as conformal variant 2. Due to the symmetry of the hexagonal lattice, the boundary conditions of the FDTD box were set to be antisymmetric along the x direction, symmetric along the y direction, and a perfectly matched layer along the z direction. The FDTD simulations were run on a Selecta Z690 desktop equipped with a liquid-cooled Intel^®^ Core 12th generation I7-12700K (12 core) and 64 GB of RAM. An average simulation time of about 3 h was required to complete a single simulation.

The geometrical parameters resulting from the optimization process, comprising the analysis of SEM images, together with the simulation outcomes, are R1 = 195 nm, R2 = 60 nm, R3 = 37 nm, R4 = 125 nm, a gold thickness of 130 nm, and a pitch of 507 nm (see Figure 1).

Several 2D frequency-domain field and power monitors were placed in crucial planar sections of the FDTD model. In particular, the x-normal and y-normal monitors were centered in the origin of the FDTD box coordinate system, while the z-normal one was placed at z = 130 nm (coinciding with the FS).

Consequently, the electric field magnitude (M) was extracted as:(1)$$M\;(x,y,z,\lambda )=\sqrt{Re(|{E_{x}(x,y,z,\lambda )|}^{2}+|{E_{y}(x,y,z,\lambda )|}^{2}+|{E_{z}(x,y,z,\lambda )|}^{2})}.$$

Finally, the electric field spectral profile at the FS was calculated by integrating Equation (1) in real space.

## 3. Results and Discussion

Figure 2 shows the measured R and T for the plasmonic grating under investigation.

The typical spectral shape of R exhibits a deep minimum set at 770 nm when n_AIR_ is considered, and 820 nm, for the case of n_EtOH_, corresponding to the main plasmonic resonance, in accordance with the position of the maximum in the T spectra. At around 700 nm, a minimum is instead placed in T, with a maximum in R. This is the fingerprint of the bandgap opening at the SiO_2_/Au interface for the folding point at the center of the Brillouin zone for the SiO_2_/Au plasmon polaritons [30].

On the smaller-wavelength side of such a folding point, other more complex structures can be seen, identifying additional mixed polaritonic and localized plasmonic modes.

When the surrounding medium is changed from n_AIR_ to n_EtOH_, an overall red shift of the R and T curves can be observed, but the bandgap remains practically in the same position. This effect is clearly evident in T, where the main plasmonic resonance moves from around 770 nm to 820 nm, with a shift corresponding to about 50 nm.

As reported in [25], the peak in T at 770 nm in air has been related to a maximum in terms of the fluorescence enhancement of the ATTO700 dye. Therefore, the observed shift is expected to have an impact on the fluorescence emission too.

Figure 3 shows the normalized fluorescence spectra of the ATTO700 dye deposited on the grating active surface as well as the ones measured on the SiO_2_ reference. As already observed in [25], in the case of air as a surrounding medium, the shape of the fluorescence emission is strongly affected by the interaction with the plasmonic grating. The plasmonic modes and fluorescence enhancement coupling effect is visible either when both excitation and collection are performed from the FS or when they are performed from the BS. When passing from n_AIR_ to n_EtOH,_ considering the absolute signal, there is no significant change apart from a general increase in the emission intensity. In the present work, we wanted to focus our attention on spectral evolution, with the normalized spectra being considered for our analysis. Thus, we want to highlight that we focused on the relative fluorescence enhancement response on the different spectral regions and not the absolute value of the enhancement factor.

Finally, in Figure 4 the ratios of the fluorescence signals measured on the plasmonic grating are reported and refer to the SiO_2_ sample for both the excitation/collection configurations (FS and BS) and the surrounding environments (air and 2% EtOH solution). For comparison, the T spectra mediated over the angles collected by the objective were also plotted. Again, the fluorescence ratios follow the shape of the T spectra, displaying a peak in air at 780 nm, corresponding to the main plasmonic resonance, consistent with the results in [25].

In the case of 2% EtOH solution as a surrounding medium, the BS and FS fluorescence ratios at wavelengths bigger than the SiO_2_/Au interface bandgap are characterized by a different behavior. In the BS configuration, a peak, in this case red-shifted at around 820 nm, is again clearly evident. On the other hand, in the FS configuration, the signal almost monotonically decreases, with no clear peaks appearing.

The largest ratio for FS configuration occurs at the lowest wavelengths, whereas the prevailing response for the BS configuration is at the highest wavelengths. This can always be observed, but the effect in solution is much more evident.

The shift in the peak positions could be related to the change in the refractive index induced by the switch from air to 2% EtOH solution.

This is consistent with [26] and can also be seen by analyzing Figure 5, where the simulated electric field in the range (500 ÷ 1100) nm is reported. The shift is clear either above or below the SiO_2_/Au bandgap region around (700 ÷ 720) nm, which remains the balance zone between the behavior of the FS and BS configurations.

Thanks to this shift, it is possible to see the peak at 680 nm within our measurement region.

However, there is an additional important effect related to the index change from n_AIR_ to n_EtOH_. Considering n_AIR_, the air/Au plasmon polaritons are located around (510 ÷ 520) nm, far from the region of interest [30], but, when changing to n_EtOH_, the H_2_O/Au bandgap (the folding point for polaritons on the FS) is shifted towards (620 ÷ 640) nm, meaning that we have to also consider the interaction with the H_2_O/Au plasmon polaritons [30].

Since, in this condition, the laser wavelength and the center of the H_2_O/Au bandgap coincide, 632 nm was the first point to be computed in terms of electric field expansion, as reported in Figure 6, where panel (a) shows the case with air and panel (d) the case with 2% EtOH solution.

Figure 6 also shows the calculations related to the electric field expansion with air and 2% EtOH solution for the peaks at 787 nm, panels (b) and (e), and 820 nm, panels (c) and (f).

In the case of the peaks at 787 nm and 820 nm, the electric field is redistributed from the BS towards the FS, when passing from n_AIR_ to n_EtOH_, but the general shape remains unchanged. This is compatible with the lower index mismatch between glass and water with respect to glass and air, which is likely also responsible for the larger measured T.

The behavior at 632 nm is more interesting.

In the case of the 2% EtOH solution, the H_2_O/Au plasmon polaritons support the electric field, enhancing its value in the range of (580 ÷ 680) nm, and are spatially bound to the H_2_O/Au interface, i.e., the FS and then the active region.

Comparing panels (a) and (d), the effect of the bandgap can be appreciated in the uncoupling of the BS and FS, resulting in an emptying of the PMMA region and suggesting the establishment of a standing wave regime.

On the other hand, the fluorescence ratio in Figure 4 exhibits a maximum at around 650 nm, between the H2O/Au bandgap and the SiO_2_/Au bandgap. Focusing on this wavelength, we calculated the electric field expansion along an x–y cross-section placed just above the active surface on the FS. The result is reported in Figure 7.

The effect of the electric field spreading along the surface due to the plasmon polaritons is evident, with an increase in the fraction of the active surface covered by the field when passing from air to 2% EtOH solution. This effect supports the increase in the electric field magnitude spectral profiles in the range of (620 ÷ 680) nm, as visible in Figure 5.

As a result, the asymmetric behavior of the fluorescence signal is similar with respect to what was proposed in [25]. The coupling of the fluorescence with the polariton modes of the SiO_2_/Au interface above 700 nm favors the directionality of the emission towards the BS. This feature is visible also in the present case but, in addition, an almost similar behavior occurs on the FS. In this case, relative to the range between (640 ÷ 700) nm, a preferential emission towards FS is observed, as shown in Figure 3 and Figure 4.

## 4. Conclusions

In the end, the synergic analysis of the simulated electric field expansions in combination with the measured optical spectra allowed for the identification of more suitable spectral regions to maximize both the dye excitation and its emission.

Consequently, this analysis provides several fundamental indications regarding: (i) the optimal choices concerning the excitation source and dye combination; (ii) the most efficient tuning of the plasmonic grating structural parameters; and (iii) the most convenient measurement (pumping/collection) scheme.

In future work, we aim to perform angle-dependent fluorescence measurements, for both the pump and the collection, to deeply understand the interplay between the plasmonic modes, while also considering their dispersion, and the fluorescence features. In addition, further analysis of the electromagnetic behavior is already planned based on more advanced methods. We aim to achieve a better representation of the scattering processes underlying the plasmonic response in the near and far field plus its interaction with the fluorophore.

## Figures and Tables

**Figure 1 materials-16-01289-f001:**
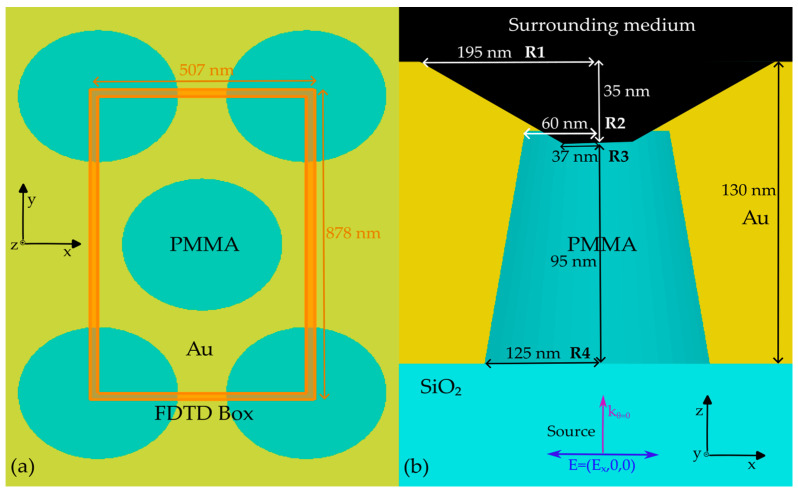
Top (**a**) and side (**b**) views of the FDTD-optimized structural model. The geometrical parameters are displayed together with their dimensions.

**Figure 2 materials-16-01289-f002:**
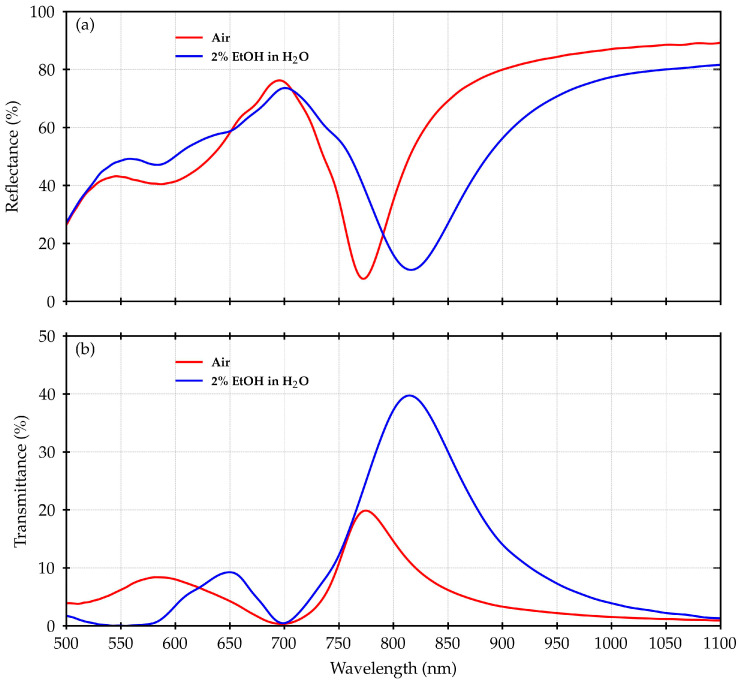
Comparison between experimental spectra acquired with the sample in air and embedded in a microfluidic cell with a 2% EtOH in water solution: (**a**) R spectra measured with an incidence angle of 2 degrees; (**b**) normal incidence T spectra.

**Figure 3 materials-16-01289-f003:**
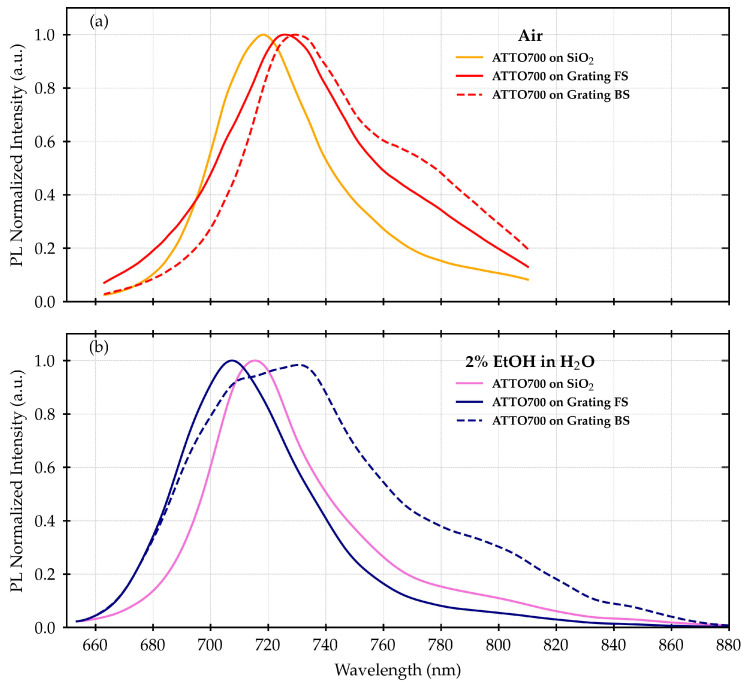
Normalized measured fluorescence spectra of ATTO700 dye deposited on the FS of the plasmonic grating and the reference SiO_2_ slide measured with the active surface exposed to air (**a**) and a 2% EtOH in water solution (**b**).

**Figure 4 materials-16-01289-f004:**
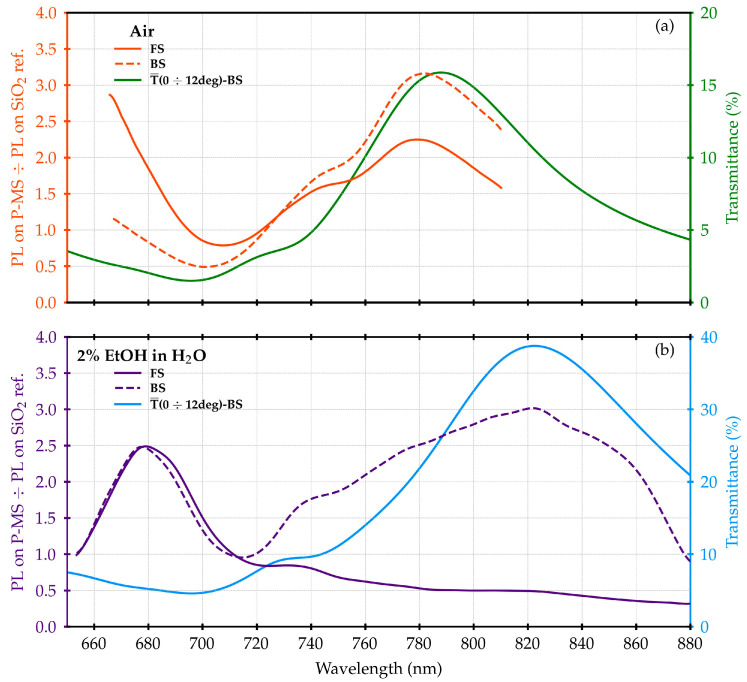
Ratios of the normalized measured fluorescence spectra of ATTO700 dye measured on the plasmonic grating with respect to the SiO_2_ reference. The curves reported in panel (**a**) represent the measurements in air for both the FS and BS configurations. In panel (**b**), the corresponding curves in 2% EtOH in water solution are shown. The T signals, mediated over the angles collected by the objective, are also plotted as a comparison.

**Figure 5 materials-16-01289-f005:**
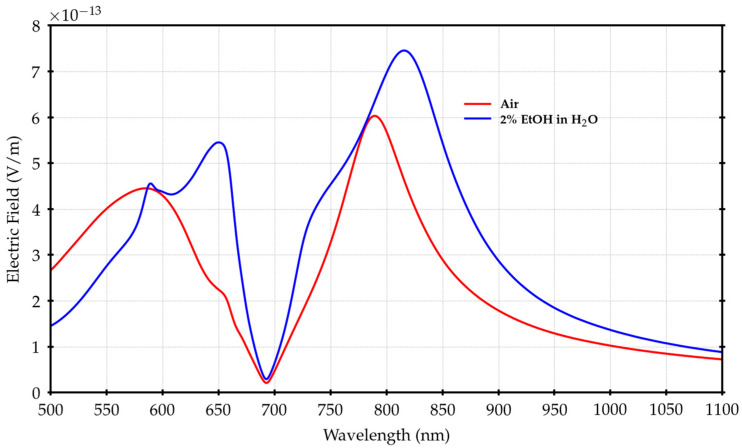
Spectral profiles of the electric field magnitude calculated by integration in the x–y horizontal plane placed at z = 130 nm for air and the 2% EtOH solution as surrounding mediums.

**Figure 6 materials-16-01289-f006:**
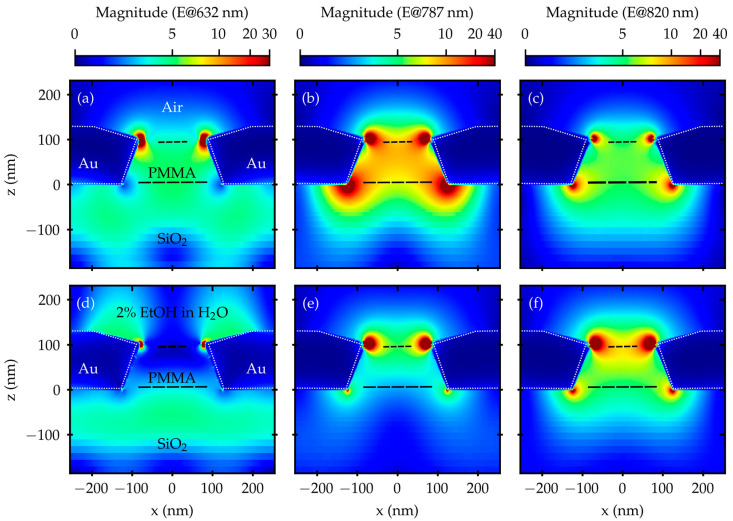
Electric field expansion calculated in the x–z plane at 632 nm, in panels (**a**,**d**), at 787 nm, in panels (**b**,**e**), and at 820 nm, in panels (**c**,**f**). For panels (**a**–**c**), the surrounding medium is air (*n* = 1.000), while in panels (**d**–**f**), it is 2% EtOH in water solution (*n* = 1.334).

**Figure 7 materials-16-01289-f007:**
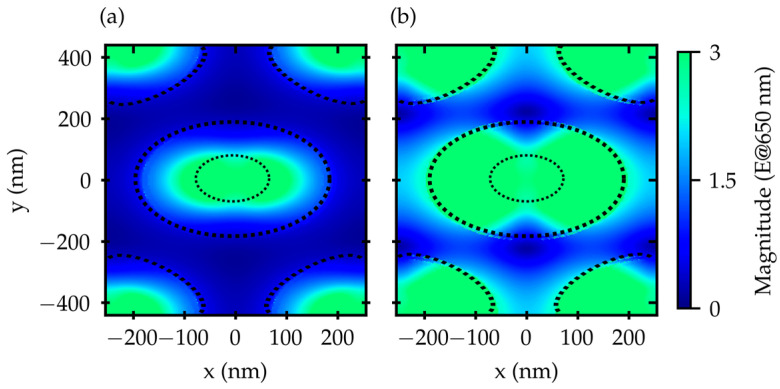
Electric field expansion calculated at 650 nm on the grating active surface along a x–y cross-section placed at z = 130 nm. The surrounding medium is air in panel (**a**) and 2% EtOH solution in panel (**b**).

## Data Availability

Not applicable.

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
