# Peer review of "Refractive Index Dependence of Fluorescence Enhancement in a Nanostructured Plasmonic Grating"

_materials, 2023, doi:10.3390/ma16031289_

Round 1

Reviewer 1 Report

A plasmonic grating is proposed for enhanced absorption and the influence of the background refractive index is investigated. Different resonances are identified accompanied with a shift of the resonance frequency and a modification of the spectral weight of the responses. The reflectivity, the transmissivity as well the photoluminescence intensity is calculated as function of operational frequency and the enhancement in the response in the presence of the grating for various host materials is represented. The spatial distribution of the electric field in the near region reveal the nature of resonances sustained for each material.

The paper is interesting but requires certain modifications before being published at MDPI Materials. In particular:

(A) The Figure A1 should be the first Figure of the paper, accompanied by a thorough description. It is difficult for the reader to understand the physical configuration and follow the included graphs.

(B) A detailed description on how the optimized parameters are selected is required. What is the dependence of the response on the most crucial design sizes?

(C) It would be interesting if the authors discussed the possibility of semi-analytical treatment of the structure with models used in solving similar setups with edges [1,2].

(D) How the results are changing when oblique incidence of electromagnetic waves are considered? The authors are considered to mention the modifications in their approach in connection with other followed methodologies [3,4]

[1] Semi-analytical model of the optical properties of a metasurface composed of nanofins, JOSA B, 2021.

[2] Rigorous analysis of a metallic circular post in a rectangular waveguide with step discontinuity of sidewalls, IEEE T MTT, 2007.

[3] Asymptotic Analysis of Transverse Magnetic Multiple Scattering by the Diffraction Grating of Penetrable Cylinders at Oblique Incidence, Journal of Applied Mathematics, 2011.

[4] Integral equation analysis of a low-profile receiving planar microstrip antenna with a cloaking superstrate, Radio Science, 2012.

Reviewer 2 Report

Authors should be reminded that there is no conclusion section in their manuscript, which is out of the common practice in scientific papers. They have then to seriously revised their document taking into account the different remarks of the reviewer report. 

Reviewer 3 Report

The manuscript titled by “Refractive Index Dependence of Fluorescence Enhancement in a Nanostructured Plasmonic Grating”, report the experiments and simulation analysis targeting the study of the reshaping of the fluorescence signal of the ATTO700 dye, and the consequent enhancement, induced by the exposure of the plasmonic grating to a solution of 2% ethanol (EtOH) in water. This works is to extend the previous analysis in Ref. 26, Nanomaterials 2022, 12, 4339 by same group of researchers. The experiment plan and results seem to be reasonable. However, except from the peak shift as expected due to the change of refractive index, the experiments results are not very impressive, does not brings out new aspect view from Ref. 26. On the other hand, the simulation shows some interesting comparison between that in the air and in the 2% EtOH in water solution. However, I do not see how these results (asymmetry) enhance the fluorescence signal of the ATTO700 dye. In a word, I did not get the “conclusion” of the experiment, “the refractive index of the surrounding medium is thus changing, with a consequent modification of the resonance conditions”. What is the benefit of the change in the refractive index of the surrounding medium? In other word, what is the goal that the authors aim to achieve?   Does this experiment show such achievement? 

Round 2

Reviewer 2 Report

The authors should normally provide a more finalized revised version of their manuscript. Besides, they should more carefully examine the title correction regarding expression "dependence in" to be corrected according to the remark I.1 of the report.

These ultimate secondary appreciations do not remove my favorable opinion on the changes the authors undertook. 

Reviewer 3 Report

The authors have responded my comments. The manuscript is revised accordingly. I think the paper can be published as its current form.
